# 🦍 GUIrilla: A Scalable Framework for Automated Desktop UI Exploration

## Abstract

Autonomous agents capable of operating complex graphical user interfaces (GUIs) have the potential to transform desktop automation. While recent advances in large language models (LLMs) have significantly improved UI understanding, navigating full-window, multi-application desktop environments remains a major challenge. Data availability is limited by costly manual annotation, closed-source datasets and surface-level synthetic pipelines. We introduce GUIRILLA [1], an automated scalable framework that systematically explores applications via native accessibility APIs to address the critical data collection challenge in GUI automation. Our framework focuses on macOS – an ecosystem with limited representation in current UI datasets – though many of its components are designed for broader cross-platform applicability. GUIRILLA organizes discovered interface elements and crawler actions into hierarchical GUI graphs and employs specialized interaction handlers to achieve comprehensive application coverage. Using the application graphs from GUIRILLA crawler, we construct and release GUIRILLA-TASK, a large-scale dataset of 27,171 functionally grounded tasks across 1,108 macOS applications, each annotated with full-desktop and window-level screenshots, accessibility metadata, and semantic action traces. Empirical results show that tuning LLM-based agents on GUIRILLA-TASK significantly improves performance on downstream UI tasks, outperforming synthetic baselines on the ScreenSpot Pro benchmark while using 97% less data. We also release MACAPPTREE [2], an open-source library for reproducible collection of structured accessibility metadata, along with the full GUIRILLA-TASK dataset, the manually verified GUIRILLA-GOLD benchmark, and the framework code to support open research in desktop autonomy.

## 1 Introduction

Understanding user interfaces (UI) through machine learning has emerged as a critical challenge in human–computer interaction. Recent advances in large language models (LLMs) have driven progress in multimodal agents for UI automation Kapoor et al. (2024); Qin et al. (2025); Cheng et al. (2024); Pawlowski et al. (2024). While training agents to navigate mobile UIs has been extensively studied Wen et al. (2024); Lee et al. (2024) thanks to abundant datasets in this domain Deka et al. (2017); Rawles et al. (2023); Wen et al. (2023), desktop automation remains constrained. Unlike mobile, desktop environments are cluttered and dynamic: small icon-based controls often encode critical meaning for task execution. Moreover, often users face overlapping windows, popups, dialogs, and system widgets. Among others, the macOS GUI presents particular challenges due to different coordinate systems and UI standards compared to other operating systems. As a result, existing multimodal benchmarks expose three structural flaws that currently set the upper bound on performance for autonomous GUI agents:

1. **Manual annotation does not scale.** Recent benchmarks Xie et al. (2024); Kapoor et al. (2024); Li et al. (2025) rely on human-designed pipelines where every task must be demonstrated, recorded, and verified by annotators. While it is important to have high-quality data in training pipelines, the process is labor-intensive and costly, limiting the scalability needed for broad, cross-domain coverage.

---

[1] https://anonymous.4open.science/r/GUIrilla-2B0F/README.md
[2] https://anonymous.4open.science/r/GUIrilla-2B0F/macapptree/README.md

2. **Single-window UIs misrepresent real usage.** Most public corpora capture only a clean snapshot of a *single* application window, whereas real users manage overlapped windows, transient dialogs, and system widgets. Prior studies Cheng et al. (2024) report that agents trained on such simplified views reach success rates near $83\%$, yet the same architectures collapse to $\approx 38\%$ when evaluated on full-desktop scenes containing multiple windows Li et al. (2025).

3. **Automated collection requires platform-specific design.** Creating diverse, high-quality datasets for GUI agents demands OS-specific expertise to navigate varying UI conventions, event handlers, and permission models. Effective automation also requires tailored engineering to reliably parse each platform's GUI. For instance, macOS lacks robust virtualization support, significantly limiting automated crawling compared to platforms like Android. As a result, it remains significantly underrepresented in large-scale datasets, e.g., macOS UIs comprise only 0.06% of all interfaces in OS-ATLAS Wu et al. (2024), and just 2.45% among all automatically collected desktop samples.

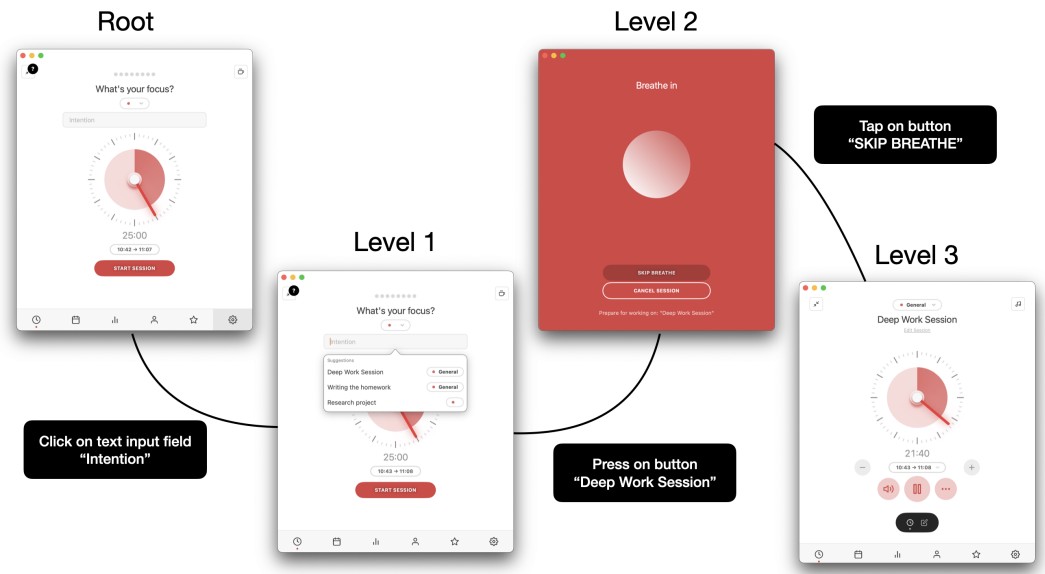

Figure 1: Parsed hierarchical tree structure from the Session application. Each node represents a UI state, containing the full accessibility tree along with a screenshot of the interface, and the edges denote GUIRILLA crawler actions. The hierarchy reflects a sequence of interactions as the agent interacts with application UI, forming the application-specific graph.

Training data has emerged as the critical bottleneck for robust desktop automation. While recent systems Gou et al. (2025) Qin et al. (2025) achieve strong benchmark performance, they do so by relying on large, closed datasets that are manually curated and not openly available. As a result, progress in the field remains gated by limited access to diverse, realistic desktop data.

To address these gaps, we introduce GUIRILLA, a fully automated framework that explores macOS GUIs at scale and summarizes them in hierarchical graph format (Figure 1). Built on macOS' accessibility API, GUIRILLA crawler systematically explores applications through simulated user interactions, supported by three GPT-4-based agents that handle meaningful element ordering, context-aware input generation, and notification of parsing obstacles.

In this work we make the following key contributions:

- **GUIRILLA framework.** The first open-source, automated framework tailored for macOS that constructs detailed full-desktop application graphs from Accessibility API snapshots and generates function-centric tasks. Application exploration utilizes specialized interaction handlers and can operate both deterministically and with LLM assistance.

- **GUIRILLA-TASK dataset.** A macOS, full-desktop corpus of 27,171 tasks across 1,108 applications and 6.8K unique screens. We also release GUIRILLA-GOLD (1,283 human-verified tasks) with a 90.26% human baseline.

- **GUIRILLA-SEE vision–language models.** We release three models: GUIRILLA-SEE (0.7B), GUIRILLA-SEE (3B), and GUIRILLA-SEE (7B). With only 6.8K images in the dataset, they exceed synthetic baselines on ScreenSpot-Pro, which were trained on vast, multi-OS datasets, showing data efficiency.

- **Open-source reproducible toolkit.** Complete end-to-end implementation including data generation pipeline, model training code, evaluation framework, and the macapptree library for collecting accessibility metadata and screenshots, facilitating reproducible automated data collection efforts on macOS.

## 2 RELATED WORK

While UI understanding has made significant progress on *mobile* Rawles et al. (2023; 2024) and *web* Liu et al. (2024b;a) platforms, largely due to the structured nature of HTML/XML and the availability of large-scale datasets (e.g., RICO) that capture visual, textual, and interactive properties across thousands of Android apps, the *desktop* setting presents unique challenges. Unlike mobile and web UIs, desktop interfaces lack a unified DOM representation and often require per-application permissions or system-level configurations for interaction and inspection. In macOS environments in particular, virtualization support is limited, making safe and scalable data collection especially difficult. As a result, automated exploration and dataset construction for desktop GUIs remains both technically challenging and relatively underexplored.

Recent progress in desktop UI grounding has been driven by the release of large-scale datasets and benchmarks. ScreenSpot Cheng et al. (2024) and its extension ScreenSpot-v2 Li et al. (2025) collected diverse task datasets paired with application screenshots across various desktop applications. ScreenSpotPro Li et al. (2025) further raised the bar by introducing high-resolution tasks in full-screen desktop environments, significantly increasing grounding difficulty. This is reflected in performance metrics: while agents reach up to 83.3% success on constrained, single-window tasks Cheng et al. (2024), performance drops sharply to 38.1% on full-screen, multi-domain tasks Li et al. (2025).

OmniACT Kapoor et al. (2024) introduced a multi-platform dataset spanning macOS, Linux, and Windows, but its scope is limited to 60 applications and websites, collected manually. OS-Atlas Wu et al. (2024) automates macOS data collection via the Accessibility API, producing a dataset of single-step question–answer pairs. However, exploration strategies remain shallow (e.g., random/depth-first search), raw accessibility label is used, and, to the best of our knowledge, the end-to-end macOS crawler code is not publicly released, which limits reproducibility of the collection process.

Also, as far as we know, existing approaches do not construct structured interaction graphs where edges represent functional tasks between interface states. Furthermore, no current OS-level crawler produces graph representations of this kind or integrates safe, agentic exploration specifically aimed at function-focused task synthesis. In contrast, our GUIRILLA framework introduces a scalable, safe, and open-source approach for dataset construction. It employs three collaborative GPT-4 agents to drive task synthesis, interface exploration, and grounded action execution. The resulting dataset, GUIRILLA-TASK, covers over 1,100 macOS applications across domains and includes more than 27,000 grounding tasks. It supports full-screen resolution and multi-window scenarios, features both screenshots and accessibility trees, and includes functionality-focused agentic task formulations. A detailed comparison with existing datasets is provided in Table 1.

| Dataset | #Apps | #Tasks | #Unique UIs | Collection | Desktop | macOS | Fullscreen | Grounding | Agentic |
|---|---|---|---|---|---|---|---|---|---|
| OSWorldXie et al. (2024) | 10 | 369 | - | Manual | ✓ | × | ✓ | × | ✓ |
| OmniACTKapoor et al. (2024) | 60 | 9802 | - | Manual | ✓ | ✓ | ✓ | × | × |
| ScreenSpot-V2Wu et al. (2024) | 6 | 324 | 187 | Manual | ✓ | × | × | ✓ | × |
| ScreenSpotProLi et al. (2025) | 23 | 1581 (511) | - | Manual | ✓ | ✓ | ✓ | ✓ | × |
| Mind2WebDeng et al. (2023) | × | 2350 | 2350 | Manual | × | × | × | ✓ | ✓ |
| OSAtlasWu et al. (2024) | - | - | 2.2M (1339) | Automated | ✓ | ✓ | ✓ | ✓ | × |
| Web-HybridGou et al. (2025) | × | - | 773K | Automated | × | × | × | ✓ | × |
| **GUIrilla-Task** | **1108** | **27171** | **6835** | Automated | ✓ | ✓ | ✓ | ✓ | ✓ |

The number in brackets denotes the reported quantity for macOS.

Table 1: Comparison of Existing Datasets for Task Automation

# 3 METHODOLOGY

GUIRILLA introduces a *graph-centric, fullscreen* exploration pipeline for macOS GUIs. Our framework builds on macOS Accessibility API [3], while integrating three specialized agents that interpret accessibility metadata, prioritize interface elements, and generate contextually appropriate actions.

## 3.1 CRAWLER

The single-app processing pipeline 2 has the following stages. First, the input bundle undergoes a standard installation routine, and together with user-specified set of parameters (such as maximal desired graph depth, and duration of parsing, the full list is available in Appendix A.2), crawler manages each of the windows of the installed app. Upon installation, the crawler attempts to extract an application's accessibility tree according to macOS accessibility framework. This framework enables simpler interaction with UI elements on the screen grouping them into a hierarchical tree structure, where each element contains essential properties such as name, role, description, position, and size. However, application developers must manually annotate or update accessibility metadata. This manual annotation process often results in error-prone accessibility trees with significant limitations: some trees contain UI elements that remain in the tree after disappearing from view, others include components with incorrect role classifications, and inaccurate positioning information.

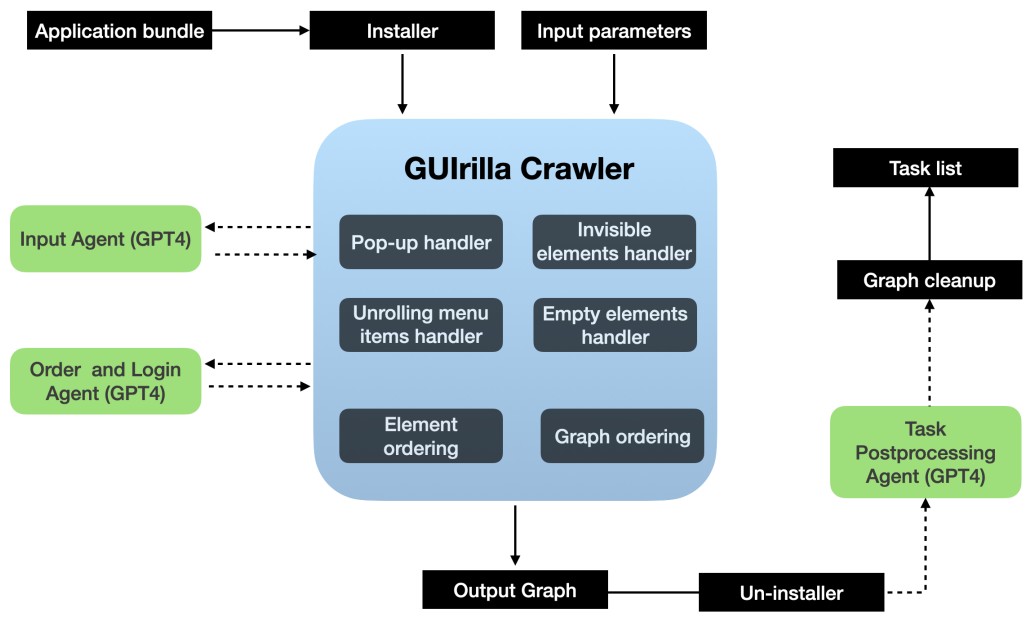

Figure 2: Architecture of the GUIRILLA framework. The GUIRILLA crawler, equipped with various UI handlers, processes an application bundle using input parameters and installer routines. It interacts with autonomous GPT-4 agents (Input, Order, and Login Agents) to navigate the application. The resulting output graph is refined by a Task Postprocessing Agent (GPT-4), which handles uninstallation and graph cleanup, ultimately producing a structured task list. The dashed line denotes the optional usage of LLMs for app exploration.

To handle these edge cases, our GUIRILLA crawler incorporates multiple specialized handlers, as shown in Figure 2. Within the crawler's core (highlighted in blue), multiple handlers address the common parsing challenges: Pop-up handler manages transient modal content, Invisible elements handler filters off-screen components present in accessibility, Unrolling menu items handler processes dynamically generated navigation elements, and Empty elements handler resolves placeholder elements with missing metadata. This multi-handler approach enables robust extraction of actionable interface information despite the underlying data quality issues.

---

[3]https://developer.apple.com/documentation/accessibility/accessibility-api

The GUIRILLA crawler performs three types of interactions to explore an application: click, cursor move, type, and press Enter key using *pyautogui* Sweigart (2015) library. To enable meaningful interaction with applications, the parsing is supported by three GPT4-based agents (the prompts are available in Appendix A.3):

1. The *Input Agent*: This agent generates contextually appropriate input strings based on the accessibility tree, ensuring relevant text is entered into form fields and search boxes.

2. The *Order and Login Agent*: Given a hierarchical list of on-screen elements, the agent determines an safest interaction sequence starting with elements that cause minimal UI changes and progressing to those with potentially significant effects (e.g., "Delete" buttons). Login pages are treated as a special case, requiring human input. This agent enhances the security and safety of the exploration process by avoiding random or destructive actions.

3. The *Task Agents*: After the uninstallation phase, these agents refine the resulting output graph, cleaning up duplicates, and transforming the structured data into a readable list of natural language tasks. Their inclusion enables both refinement and generation of more complex and natural language task descriptions.

While our framework leverages GPT-based agents to enable robust and secure interaction, both the application graphs and task data can also be collected deterministically without GPT-4 requests by following a fixed element processing order and using default input string values. However, incorporating GPT-based reasoning significantly improves the safety and contextual relevance of interactions. A detailed comparison between deterministic and GPT-guided exploration is provided in Appendix A.7.2.

## 3.2 GRAPH STRUCTURE

The application graph collected with GUIRILLA crawler consists of nodes and edges that represent application states and actions, respectively (see Figure 1). All interaction graphs are automatically annotated and visualized as accompanying SVG files. Across applications, the graphs have an average depth of 3.5, with the deepest graph reaching a depth of 101. Each node corresponds to a specific UI state of an application and contains the following fields:

- *Element*: The accessibility tree of the application window at a state.
- *Image name*: The filename of the full desktop screenshot associated with a state.
- *Actions*: A list of actions that can be executed without causing significant changes to the UI. We define a significant change as the addition or removal of more than 10 UI elements following an interaction.

Each edge captures a possible interaction and includes:

- *Action*: Information about the UI element that triggered the interaction, along with a human-readable action description and a structured dictionary representation that has a 1-to-1 map to *pyautogui* commands.
- *Out vertex*: The resulting UI state after the interaction of the crawler with the GUI.

## 3.3 TASK GENERATION

We constructed a comprehensive task dataset from the collected application graphs using a multi-stage pipeline. First, we removed redundant screenshots by filtering out actions that did not produce observable changes in the UI state. Then, we refined the initial deterministic task descriptions using our GPT-4–based Task Agent, which rewrote raw strings into more natural, function-oriented language suitable for grounding and instruction following.

To further expand the dataset, we incorporated screenshot-based task generation into the postprocessing pipeline. The Task Agent evaluates each task by considering both the target element's representation in the accessibility tree and its visual appearance in the corresponding UI screenshot. Our generative pipeline operates in two phases: (i) Click-based task generation, which focuses on user interactions with visible UI elements, (ii) Text-input task generation, which creates tasks involving keyboard input in appropriate text areas or input fields.

## 4 RESULTS

### 4.1 GUIRILLA-TASK STATISTICS AND COLLECTION PIPELINE

To construct the GUIRILLA-TASK dataset, we deployed our automated crawler on 12,298 macOS applications using an open dataset of MacAppStore apps by Sergii Kryvoblotskyi (2025). Out of these, 1,108 applications were installed, supported the macOS Accessibility framework and yielded interaction graphs. The resulting dataset spans a wide range of domains including productivity, creative tools, system utilities, and developer software, which ensure diverse coverage across common UI paradigms. The final dataset contains 27,171 tasks across 23 app genres (see Figure 4). Each task pairs a full-desktop screenshot with the corresponding accessibility tree and specifies a concrete interaction (mouse click or keyboard input). Tasks range from simple actions such as "open settings" to function-level instructions like "change your working hours to end at 18:00". Each task is classified into task type (e.g., navigation, settings) and element category (e.g., button, menu, input field).

The detailed statistics of the collected dataset, along with a representative sample and details on entry attributes are listed in Appendix A.1. We ran the crawler pipeline on a cluster of four 16 GB RAM M1 Mac Mini machines running macOS 14.7.5 Sonoma, as well as two MacBook Pros. Each machine supported parallel exploration environments per host.

### 4.2 SYNTHETIC DATA QUALITY: GUIRILLA-GOLD

To assess the reliability of macOS accessibility (AX) metadata and the quality of GPT-generated task strings, we hired 5 annotators, who were given the test split data. Annotators with accessibility expertise reviewed each data entry along five dimensions: (1) task feasibility; (2) task instruction clarity and editing for ambiguity; (3) manual task execution; (4) accessibility tree quality rating (Good/Medium/Bad scale); and (5) element-level verification of semantic properties (role, description, value) and bounding-box accuracy. Detailed annotation guidelines are provided in Appendix A.8.

**Task Quality**. From the 1319 original English language-based tasks, 84.3% of tasks were marked as DOABLE after manual verification. Comparing GPT strings to human edits, 91% required no change. The 109 edited cases showed 97% similarity to originals (Ratcliff/Obershelp), confirming minor edits. We release manully edited dataset as GUIRILLA-GOLD [4].

**AX Quality**. Accessibility metadata quality varies significantly: 64% of screens received GOOD ratings, 24% MEDIUM, and 12% BAD. At the element level, only 40% have correct role and description pairs, while 49% contain role information only, and 11% are mislabeled. Bounding boxes are accurate for 80% of elements, though 10% extend outside the visible window. This metadata sparsity and noise make accessibility-only task generation unreliable. We therefore recommend combining accessibility trees with screenshots and applying vision-based semantic adjustment to generate more precise, function-oriented, visually grounded tasks.

### 4.3 EVALUATION: GROUNDING

We fine-tune and release three GUIRILLA-SEE agents of varying parameter scales on our GUIRILLA-TASK dataset: GUIRILLA-SEE (0.7B) (based on Florence-2-large Xiao et al. (2024)), GUIRILLA-SEE (3B) and GUIRILLA-SEE (7B) (based on Qwen-2.5-VL-Instruct Bai et al. (2025)). All models are trained exclusively on GUIRILLA-TASK dataset. For training details, see Appendix A.4.

**macOS Grounding Evaluation**. We evaluate grounding by functional category on the GUIrilla-Task test set and compare against multi-OS baselines (UI-TARS, OS-Atlas, UGround). We find that across functional categories GUIrilla models achieve strong performance, with particularly large gains in Settings (+8.7), Connectivity (+26.3), Files (+7.5), Input (+8.7), as can be seen in Table 2. These categories are representative of realistic core macOS desktop tasks, that are not usually covered in web datasets that can show specific value we can bring to desktop automation. Importantly, improvements are spread across element types as well (buttons, input fields, dialogs), the full table can be found in Appendix, Table 7.

---

[4]https://huggingface.co/datasets/GUIrilla/GUIrilla-Gold/

| Model | Communication | Files | Navigation | Search & Information | E-commerce | Accounts | Input | Apps | Media | Settings | Connectivity | Total |
|---|---|---|---|---|---|---|---|---|---|---|---|---|
| UI-TARS 2B | 27.6% | 45.6% | 53.3% | 49.5% | 52.2% | 61.9% | 31.3% | 50.0% | 35.3% | 50.3% | 42.1% | 47.53% |
| UI-TARS 1.5 7B | 48.3% | 67.0% | 63.9% | 74.7% | 72.6% | **81.0%** | 56.5% | 68.8% | 54.9% | 80.9% | 68.4% | 69.07% |
| OS-Atlas 7B | 48.3% | 64.9% | 59.9% | 68.3% | 70.8% | **81.0%** | 53.9% | 66.7% | **60.8%** | 66.5% | 63.2% | 64.86% |
| UGround 2B | 51.7% | 63.0% | 60.8% | 70.0% | 69.0% | **81.0%** | 45.2% | 62.5% | 56.9% | 67.6% | 68.4% | 64.03% |
| UGround 7B | 62.1% | 67.4% | 68.7% | 75.4% | 69.0% | **81.0%** | 54.8% | 70.8% | 52.9% | 78.6% | 52.6% | 69.46% |
| GUIrilla-See 3B | 51.7% | 74.7% | 68.7% | 77.8% | 76.1% | **81.0%** | 57.4% | **72.9%** | **60.8%** | 82.7% | 73.7% | 73.48% |
| GUIrilla-See 7B | **65.5%** | **74.9%** | **70.5%** | **79.2%** | **78.8%** | **81.0%** | **65.2%** | 70.8% | **60.8%** | **87.3%** | **78.9%** | **75.59%** |

Table 2: Performance breakdown across task categories on GUIrilla-Task test set. Best performance per category shown in **bold**.

**ScreenSpot Evaluation**. Table 3 compares grounding accuracy on ScreenSpot-v2 Li et al. (2025) and ScreenSpot-Pro. ScreenSpot-v2 evaluates grounding on application screenshot and ScreenSpot-Pro contains challenging grounding tasks on full-screen desktop. While absolute comparisons are limited by differences in model architectures, training pipelines, and closed-source datasets, the table provides perspective on how dataset scale and composition affect performance. GUIRILLA-SEE (7B), trained on just 6.8K synthetic macOS screenshots, matches UI-TARS 1.5 (7B) on macOS-specific grounding (27.81% vs. 27.7%), while substantially outperforming other synthetic baselines like OS-Atlas (7B) and OS-Atlas (4B), despite using **300× less** data. Furthermore, the results on the ScreenSpot-v2 benchmark validate GUIrilla-See effectiveness. Our best model achieves a remarkable 90.33% grounding accuracy, surpassing OS-Atlas (7B), also trained on synthetic data. Compared to UGround (7B), which is trained on real web and Android UIs, GUIRILLA-SEE generalizes better across all settings, including multi-platform benchmarks. This suggests that synthetic training can compete with or exceed real-world data when structured UI diversity is high. To ensure fairness in evaluation, we made sure that there is no data leakage, details can be found in Appendix A.5.1.

| Model | Platform | Data | # Images | ScreenSpotv2 | ScreenSpotPro | ScreenSpotPro(macOS) |
|---|---|---|---|---|---|---|
| UI-TARS (7B) | Multi-OS | Real + Synthetic | ∼20M (est.) | 91.6% | 35.7% | 27.7% |
| **GUIrilla-See (7B)** | **macOS** | **Synthetic** | **6.8K** | **90.33%** | **23.66%** | **27.81%** |
| UI-TARS (72B) | Multi-OS | Real + Synthetic | ∼20M (est.) | 90.3% | 38.1% | 40.0% |
| **GUIrilla-See (3B)** | **macOS** | **Synthetic** | **6.8K** | **85.2%** | **19.17%** | **22.02%** |
| UI-TARS (2B) | Multi-OS | Real + Synthetic | ∼20M (est.) | 84.7% | 27.7% | 15.4% |
| OS-Atlas (7B) | Multi-OS | Synthetic | 2.2M | 83.3% | 18.9% | 20.0% |
| ShowUI (2B) | Real | Synthetic | 256K | 77.3% | 7.7% | 10.8% |
| UGround (7B) | Web + Android | Real | 1.3M | 76.3% | 16.5% | 12.3% |
| OS-Atlas (4B) | Multi-OS | Synthetic | 2.2M | 68.5% | 3.7% | 0.0% |
| **GUIrilla-See (0.7B)** | **macOS** | **Synthetic** | **6.8K** | **53.55%** | **7.34%** | **7.95%** |
| CogAgent (18B) | Multi-OS | Real + Synthetic | 40M | 52.8% | 7.7% | 4.6% |

Table 3: Grounding Accuracy Comparison on ScreenSpotPro (Full) and macOS Subset

Additionally, we see that full-screen supervision (compared to Gou et al. (2025)) as well as task formulation on a function level (compared to description-only as in Wu et al. (2024)) can enable better contextual grounding in realistic GUI settings. Additional analysis of model robustness across different decoding strategies and confidence intervals are provided in Appendix A.9.

**Cross-OS transfer.** Despite macOS-only training, GUIRILLA-SEE (7B) reaches 21.7% on Windows (ScreenSpot-Pro) and 27.8% on macOS, exceeding OS-Atlas 7B (12.3% / 20%) and UGround 7B (14.9% Windows). Thus, single-platform, function-level supervision does not preclude transfer and can outperform larger mixed-OS synthetic sets on challenging full-desktop scenes.

## 4.4 QUALITATIVE ANALYSIS

Analysis of 1,565 tasks across 227 applications reveals that macOS-specific training yields consistent improvements across fundamental UI interaction patterns. We identified 79 tasks where GUIrilla succeeds while all baselines fail, demonstrating strong understanding of macOS-specific paradigms: Finder-style dialogs ("Browse for movie destination folder"), System Preferences ("Edit advanced output settings"), and window management ("Close the Chat-with-Erix panel"). These success patterns validate our function-oriented approach, showing models learn what UI elements do rather than where they appear or their visual description.

**Failure Mode Analysis.** ScreenSpot-Pro evaluation reveals two key weaknesses: (1) icon-dense engineering tools such as Vivado, where tasks like "click group by repository button" or "open TCL console" fail due to limited representation of compact, icon-heavy UIs in the dataset; and (2) creative

software like Illustrator and DaVinci Resolve, where canvas-focused actions such as "draw a circle" or "select brush tool" expose insufficient coverage of creative workflows(Table 6). The model performs well on office applications and system-level tasks, suggesting macOS-focused training generalizes across typical desktop environments but requires targeted data collection for specialized professional domains. This can be mitigated by extending crawling to more creative apps and using accounts with pre-filled user-generated content in the future work, that allow for more content manipulation.

### 4.5 Evaluation: Agents

**Models.** We evaluate a range of vision-language models (VLMs) varying in size, architecture, and specialization on the GUIRILLA-TASK (agentic). These include proprietary systems like OpenAI Computer Use OpenAI (2025) and Claude Computer Use Anthropic (2024) as well as open-source models: UI-TARS 1.5 (7B), UI-TARS (2B) Qin et al. (2025), Qwen 2.5 VL (7B, 3B)Bai et al. (2025), CogAgent 9B Hong et al. (2023), and OS-Atlas Pro 7B Wu et al. (2024).

**Metrics.** We report task success rates based on action accuracy. For click tasks, success requires the predicted coordinates to fall within the target element's bounding box. Input tasks additionally require exact text matches.

**Results.** Without fine-tuning, all models struggled with input tasks (max 12.5% success), highlighting the difficulty of grounded text generation in desktop environments. OpenAI Computer Use outperformed others, achieving the highest overall success rate at 64.41%. Full results are available in Appendix A.6.2.

### 4.6 Ablation study

**Impact of Accessibility Handlers on Exploration Coverage.** Native accessibility annotations vary inconsistently across applications, creating barriers to systematic exploration. The accessibility handlers anticipate UI changes and execute meaningful interactions beyond basic clicking. Testing on three macOS applications (Stocks, Maps, Weather) across graph depth, duplicate rate, task diversity, and process time shows handlers increase task discovery by 5× in Stocks and 3× in Maps while reducing duplicates and processing time (Appendix A.7.1). These handlers target platform-agnostic problems: inconsistent element labeling, hidden components, and dynamic content. The logic transfers to other operating systems facing similar accessibility inconsistencies.

**Generative Task Agents.** We compare two training approaches: (1) deterministic accessibility metadata (`name`, `role`, `role_description`, `value`), and (2) GPT-4 task descriptions from screenshots and element crops (Table 10). Accessibility metadata often reduces to generic labels like "button" or "text" without capturing functional intent. In contrast, GPT-generated descriptions understand visual context and explicit purpose. When UI screens contain similar elements, accessibility labels create ambiguous supervision signals that hinder target identification. Florence-0.7B achieved 53.55% accuracy on GPT-generated tasks versus 40.35% on accessibility-based tasks—a 13-point gap demonstrating that functional supervision outperforms surface-level properties for training UI agents.

**Impact of backbone model.** We further examined how the choice of backbone model influences performance. When trained on our dataset, a Qwen2-VL 7B backbone already surpasses OS-Atlas, despite both using the same underlying model. Notably, OS-Atlas was trained on nearly **300×** more data, yet our model achieves higher accuracy: +1.34% in average and +2.02% on the macOS subset of ScreenSpotPro. These results highlight the *data efficiency* of our approach.

## 5 Impact, Limitations, and Ethics

**Broader Impact.** This work has significant potential to advance accessibility technology development, directly benefiting users with disabilities who rely on assistive technologies. By systematically collecting UI interaction data, our framework can improve screen readers, voice-controlled interfaces, and other adaptive technologies that help users navigate complex desktop environments.

**Technical Limitations.** Our approach is primarily constrained by dependence on developer-provided accessibility metadata, which exhibits considerable variation in quality across applications. While

currently implemented for macOS, the methodology can be adapted to other platforms such as Windows[5], Linux[6], and Android[7] by leveraging their existing accessibility infrastructures, though this requires platform-specific engineering. Additionally, solutions like OmniParser Lu et al. (2024) or Screen2AX Muryn et al. (2025) can be used to remove full reliance on accessibility metadata.

## 5.1 ETHICAL CONSIDERATIONS

We acknowledge potential risks including privacy violations, security circumvention, and malicious automation. To mitigate these concerns, we implement technical safeguards:

- **Sandboxed Environments**: We strongly recommend conducting data collection in dedicated environments with anonymized profiles to prevent accidental data leakage
- **Local-Only Operation**: All collection, replay, and annotation occur entirely locally without requiring data transmission to third parties
- **Deterministic Handlers**: Rule-based handlers enable fully offline, privacy-preserving automation without external API dependencies
- **Limited API Access**: Framework operates strictly via public macOS Accessibility APIs with no privileged system calls
- **Security-Critical Exclusion**: We explicitly avoid interaction with authentication, payment, or CAPTCHA-related interfaces

**Responsible Use Guidelines:** We explicitly discourage malicious use through clear documentation and recommend: (1) running crawlers only in controlled environments with synthetic inputs, (2) applying data filtering to remove sensitive content, and (3) using deterministic handlers for regulated data. Acceptable use cases include academic Human-Computer-Interaction research, accessibility technology development, and educational applications in controlled environments. Prohibited uses include automation of financial/healthcare systems, security circumvention, unauthorized personal data collection, and creation of tools for harassment or illegal activities. We remain committed to community oversight and transparent release practices, maintaining openness to policy revisions based on feedback to ensure responsible deployment of UI automation capabilities.

**Use of Large Language Models.** Portions of this manuscript were refined with the assistance of large language models (LLMs) for grammar and style.

## 6 CONCLUSIONS AND FUTURE DIRECTIONS

We introduce GUIRILLA, a fully automated framework that addresses the critical data scarcity challenge in desktop GUI automation. By combining accessibility-based crawling with agent-guided reasoning, our approach reduces costly manual annotation while systematically exploring full-window, multi-application environments. This directly addresses the limitations of prior work, which often focuses on narrow sets of applications or relies on manual annotation pipelines. Beyond the dataset, GUIRILLA's broader impact lies in its extensibility to other operating systems and continuous automated collection pipeline, enabling agents to adapt to evolving UI standards while significantly reducing annotation bottlenecks. Our framework lays the groundwork for developing general-purpose desktop agents by standardizing scalable task collection across diverse application domains.

**Future Work.** While GUIRILLA currently leverages built-in accessibility APIs, future extensions could integrate image-to-accessibility generation techniques to enable crawling in environments where native accessibility is limited or unavailable. Another promising direction is the development of local vision–language model (VLM) agents that actively explore applications, using reinforcement learning or related approaches to improve coverage and discover novel interaction patterns. These directions would further broaden the applicability of our framework and support more autonomous, adaptive, and scalable data collection pipelines.

---

[5] https://learn.microsoft.com/en-us/dotnet/framework/ui-automation/
[6] https://gnome.pages.gitlab.gnome.org/at-spi2-core/libatspi/
[7] https://developer.android.com/reference/android/view/accessibility/AccessibilityNodeInfo

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

## A APPENDIX

### A.1 DATASET STATISTICS

The collected tasks were split into train and test subsets, such that the applications in test did not appear in train, and test applications contained larger, more complicated accessibility trees. There are 881 applications with 25,606 entries in train and 227 applications with 1,565 task entries in test.

#### A.1.1 REPRESENTATIVE SAMPLE FROM THE DATASET

Each sample in the dataset includes the following structured fields:

- **Screen ID**: Unique identifier for the UI screen.

- **App Name**: Bundle identifier of the application.

- **Task**: Natural language description of the agent's objective.

- **Raw Action**: Deterministic textual representation of the user action.

- **Action**: Structured action format, e.g., `"left click, (x, y)"`.

- **Element Data**: JSON metadata of the target UI element extracted from the accessibility tree.

- **Scaling Factor**: Display scaling factor for the specific screen.

- **Original Task**: Boolean indicating whether the task was derived directly from the original interaction graph.

- **A11y Path**: Full accessibility tree before the action was taken.

- **Image**: Full-screen desktop screenshot, stored as a binary image.

- **Cropped Image**: Subregion of the full screenshot containing the target application (variable dimensions).

- **Segmented Image**: Screenshot of the application window with segmented UI regions.

- **Task Category**: One of 22 predefined task categories (e.g., *Search & Information*, *Files*).

- **Element Category**: One of 16 UI element types (e.g., *Slider*, *Button*).

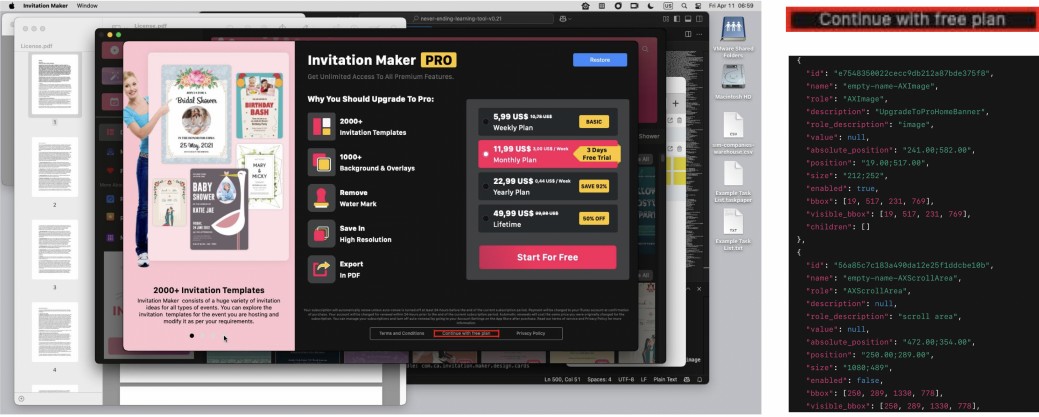

Figure 3: Example sample from our dataset. Left: screenshot of the macOS desktop interface. Upper right: target element cropped. Lower right: a segment of the accessibility tree.

### A.1.2 COLLECTION STATISTICS

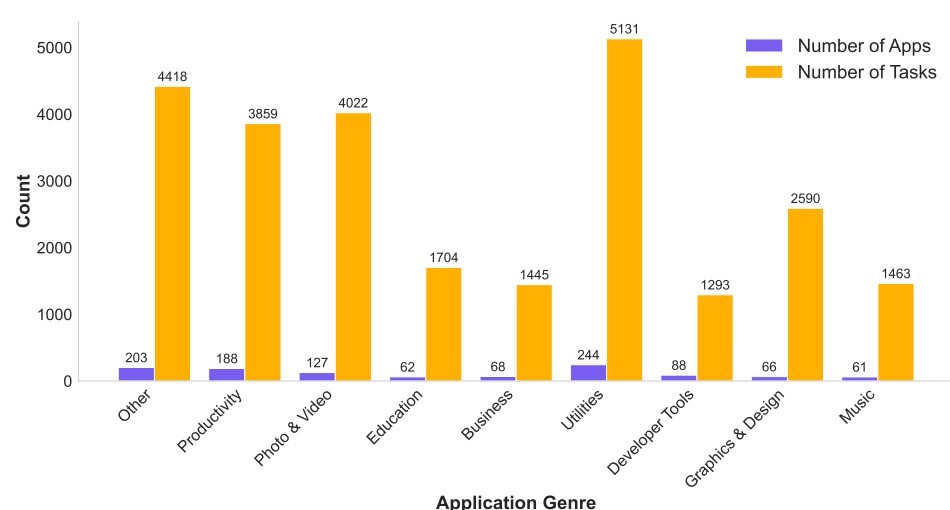

Figure 4: Number of apps and associated tasks per genre. For each genre, the left bar shows the number of apps, and the right bar shows the number of tasks. Colors distinguish between the two quantities. The figure highlights disparities between app availability and task density across categories.

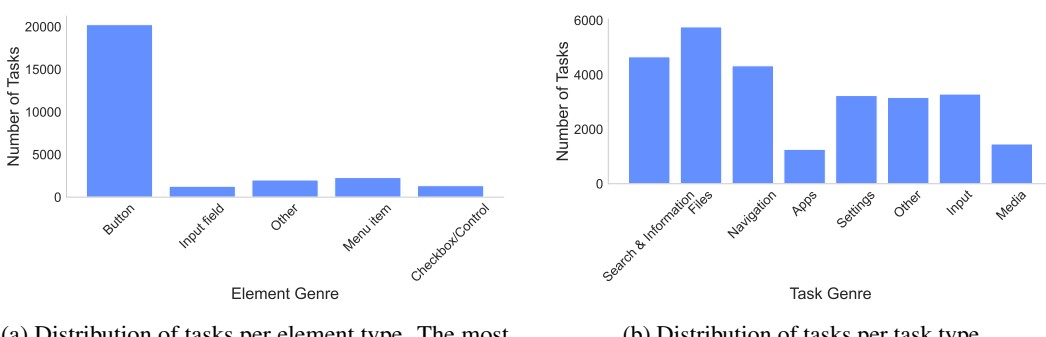

(a) Distribution of tasks per element type. The most prevalent category is buttons.

(b) Distribution of tasks per task type.

Figure 5: Distributions of tasks across element types (left) and task types (right).

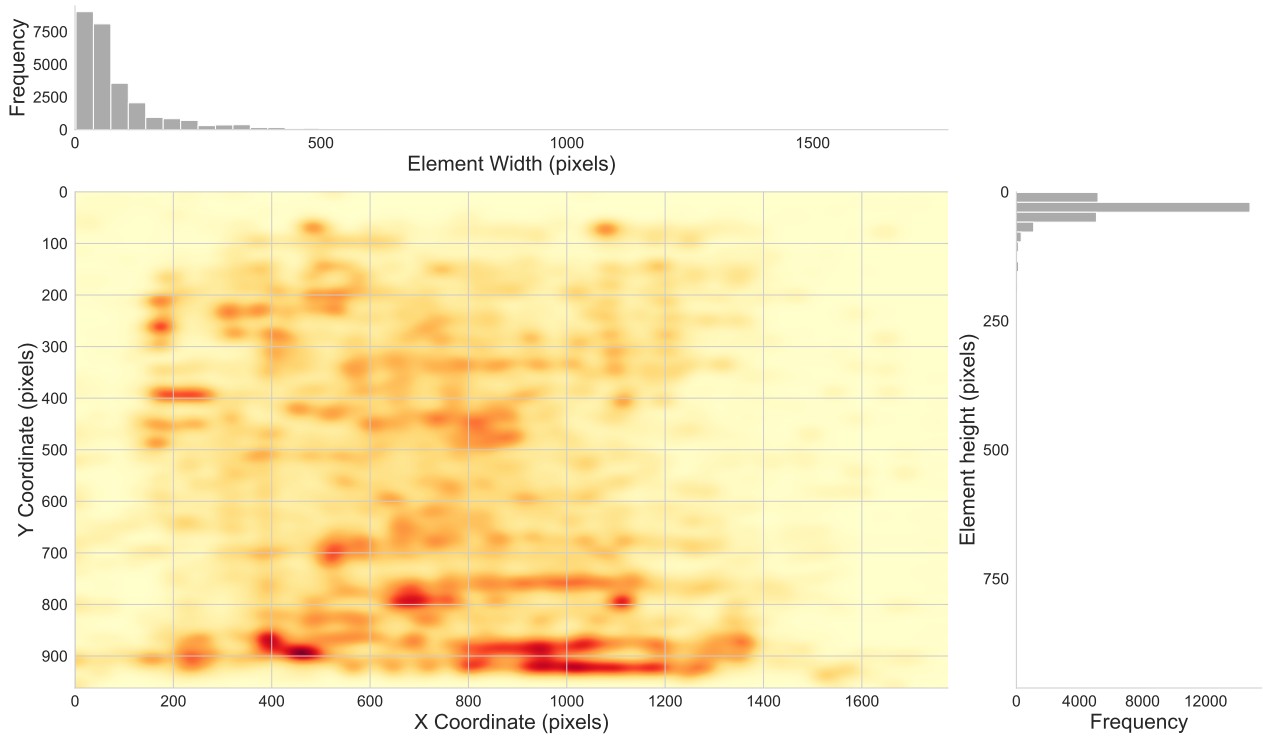

Figure 6: Top-left: Distribution of target element widths. Bottom-left: Distribution of target element center locations, showing that most target elements are positioned near the bottom of the desktop interface. Bottom-right: Distribution of target element heights.

Many interaction targets are located in peripheral regions (e.g., toolbars, corners), and a large proportion are visually small, with limited surface area (Figure 6).

## A.2 PARAMETERS

| Parameter | Description |
|-----------|-------------|
| Maximum parsing duration | Specified in minutes, default is 2 hours |
| Deterministic text input | Default string is `'DEFAULT'` |
| Maximum parsing tree depth | Default is 25 |
| Cursor move before click | Defaults to `False` |
| Agent usage | Set to `True` by default. Can be enabled if an OpenAI API key is provided in a separate file |
| Task collection | Defaults to `True`. If set to `False`, graphs can be collected without their associated tasks |

Table 4: Configuration Parameters for the Crawler

## A.3 PROMPTS

---

**Input Agent Instructions**

Analyze the given macOS application accessibility screen information and follow these steps:

1. Determine the type and purpose of the application based on the provided elements and descriptions.

2. Identify all `AXTextField` elements present in the structure.

3. For each `AXTextField`:

   (a) Infer its specific purpose within the application context.

   (b) Consider what a user would input in this field based on accessibility cues and typical behavior.

   (c) Generate an example input relevant to the field's likely function and the app's overall purpose.

**Output:** A JSON object where:

- **Keys** = integer IDs of the `AXTextField` elements
- **Values** = realistic example inputs, based on screen context

Only return the JSON object—no additional explanations.

**Examples:**

- `{7: "Yellow Submarine"}`                              // Music app search
- `{12: "John", 15: "Smith", 21: "07580198241"}` // Contacts app
- `{8: "main"}`                              // IDE project file search

**Note:** Ensure that inputs are app-appropriate and reflect common human interactions.

---

**Order Agent Instructions**

Given accessibility screen info, organize UI elements in logical interaction order. Consider irreversible actions and screen transitions.

**Output:** JSON with nested groups (max 8), each containing element IDs:

- Prioritize elements in popovers, content switches, and window controls.
- Derive element type from description if needed.
- Include **ALL element IDs** from input.

**Grouping Rules:**

- `dynamic_TYPE` — dynamic lists (emails, notes, etc.)
- `repeated_TYPE` — options where only one is needed (date, category, etc.)
- Avoid grouping unrelated or static UI items together.

**Flags to include when relevant:**

- `"login_page": true`
- `"system_access_required": true`

**Example 1 — Complex App:**

```
{
  "action_order": [
    {"menu_buttons": [1, 2, 3]},
    {"dynamic_emails": [4, 5, 6, 7]},
    {"repeated_time_selection": [8, ..., 31]},
    {"popover_buttons": [32, 33]}
  ],
```

```
  "login_page": false,
  "system_access_required": false
}
```

**Example 2 — Login Page:**

```
{
  "action_order": [
    {"login_elements": [1, 2, 3]},
    {"account_settings": [4]}
  ],
  "login_page": true,
  "system_access_required": false
}
```

## Click Task Prompt

You are given a UI screenshot, an image of the clicked UI element. The clicked element is highlighted in red. Your task is to describe the action needed to click this element.
Guidelines:
0. If the element is not perfectly selected (ex. partially), the box is strangely located, or no human would do this task - return empty string.
1. The task must describe the function, not the appearance of the element. For example, prefer "Create a new document" over "Click the grey + button." Repeating the element's text is acceptable.
2. The task must be unique to this screen. For example, if there are two buttons labeled "Open," you must specify which "Open" button is meant.
3. The task must consider the app context, but not imagine extra information. For example, if the app is an image editor and the button is "Delete," the better task is "Delete an image", not just a generic "delete."
4. Use the fewest words possible without sacrificing clarity.
5. Write the task in straightforward English only.
6. Select a category for each task. Must be one of Navigation (go back), Settings (adjust volume), Files (save file), Apps (open edge), Search & Information (check weather), Media (play music), Accounts (sign in), Communication (share file), Input (enlarge font), Connectivity (connect wifi), Modes (dark mode), E-commerce (add to cart)
7. Select a category for each element. Must be one of Image, Text, Checkbox/Control, Menu item, Input field, Button, Group, Link.
Important notes:
The click is based on accessibility information. Metadata may be incorrect or the element may not exist. Rely primarily on the images.
The element image should show a single element with a unique function. If the element is obstructed, covered by a window or pop-up, or if multiple cropped elements are shown — return an empty string.
Inspect the red box carefully: if the element is not visible, return an empty string.
If there is no red box - return empty string. Return your answer in JSON format, with no extra text.
Example:

```
{
    "task": "Open the menu to see tutorials",
    "task_category": "Search & Information",
    "element_category": "Button"}
```

## Input Task Prompt

**You are given:**

- An original task description for a UI interaction: `{task_string}`
- A screenshot showing the full interface with a red-highlighted element
- A cropped view focusing on just the highlighted element

**Your goal:** Change the task into a natural-language instruction **fully in English** that involves only inputting text. Output an action as `"type"` + the exact text to input. If not clearly solvable from the task, revise it.
**Key Principles:**

- Make it sound like a real instruction a person would give
- Use exact input (no placeholders); don't interpret content—be explicit
- Focus on real-world intent and what a user is likely trying to do

**Requirements:**

- Instruction must be clear, natural, and concise
- Action must start with `type` and include exact text
- Both fields must be fully in English
- No placeholders like "your name" or "email"
- Avoid click/press/select – only typing
- Must be obvious what to type from the instruction
- Never add phrases like "by typing it"

**Output Format (JSON):**

```
{"task": "Use john.doe@example.com as your login email",
 "action": "type john.doe@example.com"}
```

**Bad vs Good Examples:**

- "Enter coded message" → "Enter 1234 as your coded message"
- "Save your converted files..." → "Use /Users/yourname/Desktop as your destination folder"
- "Check the box labeled..." → "Select Include borders and shadings as your option"

**Avoid These Mistakes:**

- Placeholder text: "your name" → "Maria Garcia"
- Mechanical: "password in field" → "Use TrustNo1 as your password"
- UI-only focus: "Fill search box" → "Find information about electric cars"
- Vague: "Type the code" → "Enter 8294 as your verification code"
- Impersonal: "Input required" → "Add your birthday as 03/15/1988"

## A.4 TRAINING SETUP

We training 3 GUI agents on the collected dataset.

### A.4.1 GUIRILLA-SEE-0.7B

*GUIrilla-See-0.7B* is built on FLORENCE 2-LARGE ($\approx 0.7$ B parameters) and fine-tuned via supervised fine-tuning for *open-vocabulary detection* in GUI screenshots. Given an image and a free-form textual query, the model predicts either a bounding box or a polygon mask that encloses the best-matching UI element.

**LoRA configuration.** Fine-tuning uses Low-Rank Adaptation with RSLoRA initialisation:

- rank $r = 8$
- scaling $\alpha = 16$
- dropout $= 0.05$
- bias $= none$
- target modules $= \{\texttt{q\_proj}, \texttt{o\_proj}, \texttt{k\_proj}, \texttt{v\_proj}, \texttt{linear}, \texttt{Conv2d}, \texttt{lm\_head}, \texttt{fc2}\}$
- weight init *Gaussian*

**Training setup.**

- Hardware: $1 \times$ NVIDIA A100 40 GB.
- Batch size: 8, mixed precision.

- Optimiser: AdamW, learning rate $2 \times 10^{-5}$. Cosine decay schedule with a 5% warm-up fraction.
- Epochs: 4; total wall-clock time $\approx 10$ hours.

### A.4.2 GUIRILLA-SEE-3B

*GUIrilla-See-3B* starts from QWEN-2.5-VL-3B-INSTRUCT (3 B parameters) and is fine-tuned with supervised fine-tuning (SFT) to localise macOS GUI elements. Given a full-desktop screenshot and a natural-language instruction, the model outputs a single coordinate $(x, y)$ that lies at (or very close to) the centre of the referenced region.

**LoRA configuration.** Fine-tuning uses Low-Rank Adaptation (LoRA) in `bfloat16` mixed precision:

- rank $r = 32$
- scaling $\alpha = 16$
- dropout $= 0.1$
- bias $=$ *none*
- target modules $= \{\texttt{down\_proj}, \texttt{o\_proj}, \texttt{k\_proj}, \texttt{q\_proj}, \texttt{gate\_proj}, \texttt{up\_proj}, \texttt{v\_proj}\}$
- weight init *Gaussian*

**Training setup.**

- Hardware: **2 $\times$ NVIDIA H100 80 GB**.
- Global batch size: 16
- Optimiser: AdamW with $\beta_1 = 0.9,\ \beta_2 = 0.95$.
- Learning rate: $2 \times 10^{-5}$, cosine decay schedule, warm-up ratio 0.05.
- Attention kernel: **FlashAttention-2** for memory-efficient training.
- Epochs: 2; total wall-clock time $\approx 5$ hours.

### A.4.3 FINE-TUNING IMPROVEMENT ON BASE MODELS.

| Model | Base Acc. (%) | Tuned Acc (%) |
|---|---|---|
| Florence Large (0.7B) | 8.31 | 53.55 |
| Qwen 2.5 VL (3B) | 18.40 | 73.48 |
| Qwen 2.5 VL (7B) | 35.78 | **75.59** |

Table 5: Accuracy improvements after fine-tuning on GUIRILLA-TASK.

### A.4.4 GUIRILLA-SEE-7B

We also train a larger model that starts from QWEN-2.5-VL-7B-INSTRUCT (7B parameters). All LoRA, optimiser, and scheduler settings are kept *identical* to the 3B run. Using the same $2 \times$ H100 80 GB configuration with FlashAttention-2, training finishes in roughly 6–7 hours.

### A.5 SCREENSPOT DETAILS

### A.5.1 DATA LEAKAGE ANALYSIS ON SCREENSPOT

We manually screened overlaps by bundle IDs and application names to ensure no data leakage happened during training for both ScreenSpot v2 and ScreenSpot-Pro benchmarks. As ScreenSpot-v2 doesn't provide this information, we manually labeled the apps there.

We discovered the following overlaps out of 881 applications in our train dataset and ScreenSpot test sets: OneNote appears in train data (macOS app) and in ScreenSpot-v2 (Windows). This app

has 1 task in the benchmark, and the login screen looks identical, so the leakage may have affected the result. We adjusted the score to account for it from 90.41% -> 90.33%. This doesn't influence the ranking, yet we adjusted the score for fairness. Microsoft Excel appears in the train dataset and in ScreenSpot-Pro. Here we manually looked at every screen (screen ids 4650-4659) and found that our data only includes a login flow and never actually opens the main app and its functionality. In ScreenSpot-Pro on the other hand, all tasks focus on Excel functions as part of the multi screen window. So, we assume that no major leakage was done here.

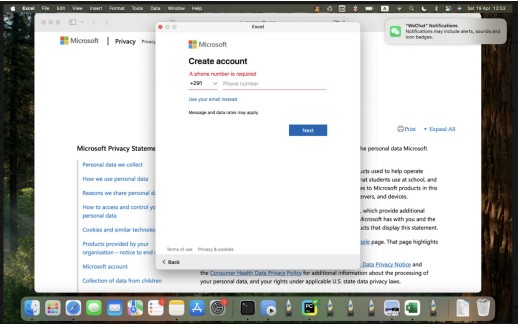
(a) GUIrilla: Excel login page.

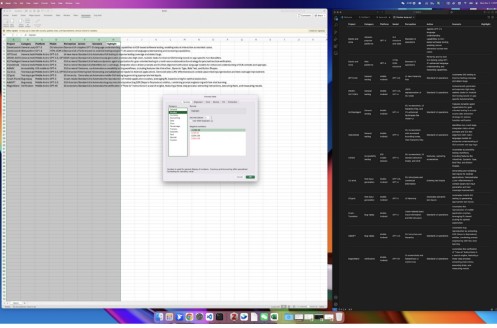
(b) ScreenSpotPro: Main app with table manipulation.

Figure 7: Side-by-side comparison of Excel app data across datasets.

| Model | Development | Creative | CAD | Scientific | Office | Overall Acc |
|---|---|---|---|---|---|---|
| UI-TARS-72B | 40.8 | 39.6 | 17.2 | 45.7 | 54.8 | 38.1 |
| UI-TARS-7B | 36.1 | 32.8 | 18.0 | 50.0 | 53.5 | 35.7 |
| UI-TARS-2B | 26.4 | 27.6 | 14.6 | 39.8 | 42.6 | 27.7 |
| GUIrilla-See-7B | 23.08 | 14.37 | 16.09 | 35.83 | 37.39 | 23.66 |
| GUIrilla-See-3B | 19.40 | 10.56 | 14.94 | 29.13 | 30.00 | 19.17 |
| OS-Atlas-7B | 17.7 | 17.9 | 10.3 | 24.4 | 27.4 | 18.9 |
| UGround-7B | 14.7 | 17.0 | 11.1 | 19.3 | 27.0 | 16.5 |
| CogAgent (18B) | 8.0 | 5.6 | 6.1 | 13.4 | 10.0 | 7.7 |
| ShowUI (2B) | 9.4 | 5.3 | 1.9 | 10.6 | 13.5 | 7.7 |
| GUIrilla-See-0.7B | 6.69 | 6.45 | 3.83 | 12.20 | 8.26 | 7.34 |
| OS-Atlas-4B | 3.7 | 2.3 | 1.5 | 7.5 | 4.8 | 3.7 |

Table 6: Task category performance per app category on ScreenSpot-Pro.

## A.6    ADDITIONAL EVALUATION DETAILS

### A.6.1    GROUNDING

In grounding evaluations (Table 7), GUIRILLA-SEE (7B) achieved the highest overall accuracy at 75.59%. GUIRILLA-SEE (7B) showed particularly strong results on buttons, input fields, and "Other" elements such as icons and links, demonstrating robust performance across varied UI components. Interestingly, UGround shows marginal advantages in menu-heavy tasks, and we found their data to contain 400× more menu samples, reflecting the limits of scale without functional task diversity.

| Model | Button | Input | Menu | Checkbox | Other | Overall |
|---|---|---|---|---|---|---|
| | | | **Grounding Accuracy (%)** | | | |
| Qwen 2.5 3B | 8.0 | 0.0 | 0.0 | - | – | - |
| Qwen 2.5 7B | 36.49 | 30.36 | 46.0 | 44.68 | 24.74 | 35.78 |
| UI TARS 2B | 50.56 | 25.0 | 52.67 | 59.57 | 36.84 | 47.53 |
| UGround v1 2B | 64.26 | 48.21 | 79.33 | 68.09 | 58.95 | 64.03 |
| OS-Atlas-Base-7B | 65.76 | 53.57 | 72.67 | 57.45 | 62.11 | 64.86 |
| UI TARS 1.5 7B | 68.57 | 64.29 | 86.67 | **78.72** | 58.42 | 69.07 |
| UGround v1 7B | 68.67 | 56.25 | **88.67** | **78.72** | 64.21 | 69.46 |
| **GUIrilla-See (3B)** | 74.48 | 61.61 | 81.33 | **78.72** | 66.84 | 73.48 |
| **GUIrilla-See (7B)** | **76.55** | **66.07** | 86.67 | 76.6 | **66.84** | **75.59** |

Table 7: Grounding accuracy across models and element categories on GUIRILLA-TASK.

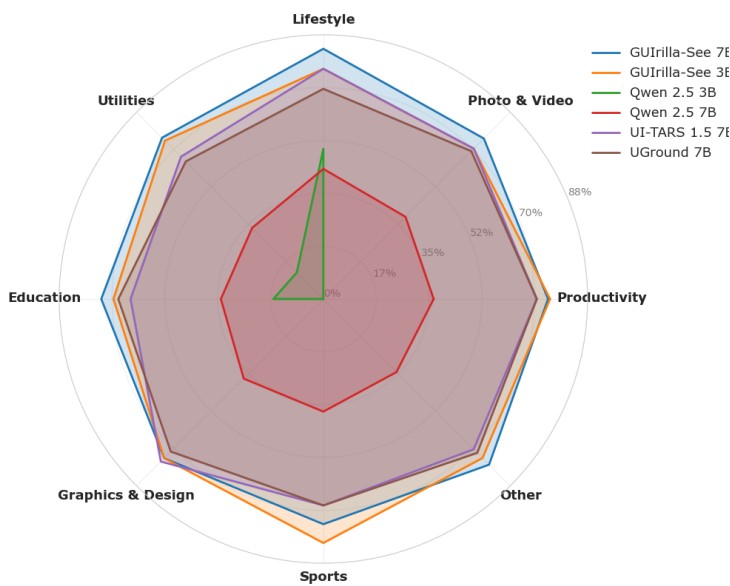

Figure 8: Comparative Performance on GUIrilla-Task (Grounding) of Vision Language Models Across Application Domains. Larger models (7B) generally outperform smaller ones, with the biggest gains in Developer Tools, Productivity, and Graphics & Design. GUIrilla-See 3B shows strong performance relative to 7B models, indicating effective domain specialization.

### A.6.2    AGENTIC

| | Success Rate (%) | | |
|---|---|---|---|
| **Model** | Input | Click | Overall |
| OpenAI Computer Use | 8.04 | **68.75** | **64.41** |
| Claude Computer Use | 8.93 | 65.59 | 61.53 |
| OS-Atlas-Pro-7B | 7.14 | 62.84 | 58.85 |
| UI TARS 1.5 7B | 1.79 | 54.65 | 50.86 |
| UI TARS 2B | 7.14 | 50.24 | 47.16 |
| Qwen 2.5 VL 3B | **12.5** | 42.95 | 40.77 |
| Qwen 2.5 VL 7B | 2.68 | 39.16 | 36.55 |
| CogAgent 9B | 3.57 | 15.83 | 14.95 |

Table 8: Success rate across models and interaction categories on GUIRILLA-TASK (agentic)

## A.7 ABLATION DETAILS

### A.7.1 INFLUENCE OF HANDLERS

| Application | Metric | Handler-Supported Crawler | Random Crawler |
|---|---|---|---|
| Stocks | Graph Depth | 14 | 16 |
| | Number of Tasks | 162 | 32 |
| | Duplicate rate | 0.08 | 0.14 |
| | Parse Time (hh:mm:ss) | 00:20:51 | 00:29:35 |
| Maps | Graph Depth | 6 | 9 |
| | Number of Tasks | 107 | 36 |
| | Duplicate rate | 0.1 | 0.2 |
| | Parse Time (hh:mm:ss) | 00:21:00 | 00:25:35 |
| Weather | Graph Depth | 6 | 7 |
| | Number of Tasks | 73 | 73 |
| | Duplicate rate | 0.0 | 0.01 |
| | Parse Time (hh:mm:ss) | 00:10:56 | 01:05:48 |

Table 9: Comparison of Handler-Supported Crawler vs Random Crawler Across Applications

### A.7.2 COMPARING DETERMINISTIC AND GPT-REFINED TASK DESCRIPTIONS

Table 10: Examples of Task Agent and Deterministic Instructions by App

| App | Task Agent | Deterministic |
|---|---|---|
| Prayer Notes | Access Prayer Notes support page | button |
| GoProPlayer | Open a media file | button Open_Media... |
| Fax | Add files or images to the fax | ADD FILES OR IMAGES button |

## A.8 GUIRILLA-GOLD DATASET ANNOTATION GUIDELINES

### A.8.1 OVERVIEW

This document provides instructions for annotators to evaluate and improve UI task datasets, with focus on accessibility principles.

### A.8.2 TASK STRING FEASIBILITY EVALUATION

**Evaluation Steps:**

- **Step 1:** Evaluate clarity and readability of the task string. Edit if ambiguous or poorly phrased.
- **Step 2:** Assess executability. Mark as DOABLE if the task is clear and the required element is visible. Mark as NOT DOABLE if the element is not visible or does not exist.

**DOABLE Examples:** "Click the Submit button", "Type 'hello world' in the search field"
**NOT DOABLE Examples:** "Click the button" (when multiple buttons present), "Enter your password" (if no password input visible)

### A.8.3 TASK EXECUTION GUIDELINES

Attempt to execute the task exactly once to verify correctness:

- **Click Actions:** Locate the correct element and click once within its bounding box
- **Type Actions:** Find the input field and type the exact text provided (case-sensitive)
- **Multi-step Tasks:** Mark as NOT DOABLE if requiring multiple distinct actions

**Constraints:** Attempt only once, no retries. Do not fabricate actions not in the task string.

### A.8.4 ACCESSIBILITY QUALITY RATING (1–3 SCALE)

**Score 1 – BAD:** Critical issues severely impact assistive technology—missing labels, incorrect roles, invisible elements, broken grouping, no logical structure.

**Score 2 – MEDIUM:** Moderate issues present—incomplete/generic labels, occasional role mismatches, partial grouping, minor positioning issues.

**Score 3 – GOOD:** Accessibility tree accurately represents visual UI—descriptive labels, accurate roles, proper grouping, logical structure, complete state information.

### A.8.5 LABEL AND ROLE VERIFICATION

Review each element's semantic description and role. Uncheck "Semantic" if the meaning or AX role is incorrect. Uncheck "BBox" if the bounding box doesn't match the visible element.

### A.8.6 QUALITY ASSURANCE CHECKLIST

☐ Task string evaluated and edited for clarity

☐ NOT DOABLE marked for invisible elements

☐ Task execution attempted once

☐ Accessibility score reflects usability

☐ Labels and roles verified

☐ Checkboxes unchecked for mismatches

## A.9 CONFIDENCE INTERVALS AND SENSITIVITY TO DECODING STRATEGIES

To provide uncertainty estimates and strengthen the reliability of model comparisons, we conducted additional experiments examining performance variability across different decoding strategies.

### A.9.1 EXPERIMENTAL SETUP

Our primary results were obtained using greedy decoding with the following generation parameters: `num_beams=3`, `do_sample=False`, `temperature=None`, `top_p=None`, `top_k=None`.

To assess performance variability, we re-evaluated all models using stochastic decoding with parameters: `num_beams=3`, `do_sample=True`, `temperature=0.3`. For each model, we conducted three independent runs and computed mean ± standard deviation of success rates.

### A.9.2 RESULTS

Table 11 presents results for both decoding strategies on the GUIrilla-Task test set.

| Model | Greedy (%) | Sampled (T=0.3) (%) |
|---|---|---|
| Florence Base | 10.73 | 10.64 ± 0.10 |
| Florence Large | 8.31 | 8.08 ± 0.10 |
| Qwen 2.5 VL 3B | 17.96 | 18.06 ± 0.22 |
| Qwen 2.5 VL 7B | 35.78 | 35.40 ± 0.45 |
| UI-TARS 2B | 47.54 | 18.43 ± 0.22 |
| UI-TARS 1.5 7B | 69.07 | 52.17 ± 0.42 |
| UGround v1 2B | 64.03 | 64.41 ± 0.00 |
| UGround v1 7B | 69.46 | 69.84 ± 0.06 |
| OS-Atlas-Base-7B | 64.86 | 61.82 ± 0.10 |
| GUIrilla-See-0.7B | 53.55 | 53.48 ± 0.32 |
| GUIrilla-See-3B | 73.48 | 73.90 ± 0.03 |
| GUIrilla-See-7B | 75.59 | 75.85 ± 0.06 |

Table 11: Performance comparison between greedy and stochastic decoding strategies. Models fine-tuned on GUIrilla-Task (bottom section) show consistent performance with minimal variance, while some pretrained models exhibit notable degradation under stochastic decoding.

The results reveal distinct patterns in decoding sensitivity. Fine-tuned GUIrilla-See models demonstrate remarkable consistency across decoding strategies, with standard deviations below 0.32% in all cases. This stability suggests robust learning of UI interaction patterns.

In contrast, several pretrained models show significant performance degradation under stochastic decoding, most notably UI-TARS models which experience drops of 16-29 percentage points. This sensitivity highlights the importance of decoding strategy selection and suggests that some models may be overfitting to specific generation patterns during pretraining.

The minimal variance observed in our fine-tuned models provides confidence in the reported performance gains and demonstrates the robustness of the GUIrilla training approach across different inference conditions.

