# OpenReview forum: "GUIrilla: A Scalable Framework for Automated Desktop UI Exploration"
_ICLR.cc/2026/Conference — ICLR 2026 Conference Withdrawn Submission_

### Official Review · Reviewer_HRuy · 2025-10-29

**Soundness:** 3
**Presentation:** 3
**Contribution:** 4
**Rating:** 6
**Confidence:** 4

**Summary:**

The paper presents GUIRILLA, a scalable, automated framework for exploring desktop GUIs on macOS and generating function-grounded tasks. The system crawls apps via the macOS Accessibility API and builds hierarchical GUI graphs where nodes are UI states and edges are actions (click, move, type, enter). Specialized handlers address platform-specific AX noise (pop-ups, invisible/empty elements, dynamic menus), and three GPT-4 agents prioritize safe interaction, generate contextual inputs, and rewrite tasks into concise, function-level language. Using this pipeline, the authors release GUIRILLA-TASK: 27,171 tasks across 1,108 macOS apps with full-desktop screenshots and AX trees, plus GUIRILLA-GOLD (1,283 human-verified tasks) and MACAPPTREE, an open-source metadata collection library. They fine-tune three VLMs (0.7B/3B/7B) that achieve strong macOS grounding vs. multi-OS baselines on their test set and competitive results on ScreenSpot-v2 and the macOS subset of ScreenSpot-Pro using ≈300× less training data than UI-TARS/OS-Atlas. Ablations show handlers notably increase task discovery and GPT-based task phrasing yields better supervision than raw AX labels. Limitations include reliance on AX quality, limited action types, and weaker performance on icon-dense professional/creative tools.

**Strengths:**

- Originality: Introduces a macOS-focused, open-source crawling framework that organizes exploration into hierarchical GUI graphs and leverages targeted handlers for AX inconsistencies (pop-ups, hidden/empty elements, dynamic menus) (Sec. 3.1–3.2; Fig. 2). The function-level task generation from both AX and screenshots is a useful formulation (Sec. 3.3).
- Quality: Comprehensive dataset construction at scale (1,108 apps; 27k tasks) with full-desktop screenshots and AX trees (Sec. 4.1; Fig. 3–5). Human validation (GUIRILLA-GOLD) shows 84.3% task feasibility and high agreement with GPT rewrites; detailed AX quality analysis highlights noise and motivates multi-modal grounding (Sec. 4.2).
- Empirical support: Solid macOS grounding gains across categories vs. multi-OS baselines (Table 2); competitive performance on ScreenSpot-v2 and macOS subset of ScreenSpot-Pro with far less data (Table 3). Clear ablations on handler benefits (Table 9) and the value of functional, GPT-refined supervision (Table 10). Data leakage analysis is provided (Sec. A.5.1).
- Clarity and reproducibility: Pipeline is clearly illustrated (Fig. 1–2), prompts are released (App. A.3), and the MACAPPTREE toolkit plus dataset/model releases support reproducibility (Abstract; Sec. 4.1).
- Significance: Addresses the underrepresented macOS ecosystem and full-desktop, multi-window scenarios, a known gap limiting GUI agents (Sec. 1–2). Demonstrates data efficiency and cross-OS transfer (Sec. 4.3).

**Weaknesses:**

- Limited action diversity and task complexity: The crawler models mostly single-step clicks and text input; no explicit support for drag/scroll/right-click/keyboard shortcuts or multi-step, compositional flows (Sec. 3.1; 3.3). Graphs have shallow average depth (3.5), suggesting limited coverage of complex workflows (Sec. 3.2).
- AX dependency and heuristics: Despite handlers, exploration and state change detection hinge on noisy AX metadata and a heuristic threshold for “significant changes” (±10 elements) (Sec. 3.2), which may miss subtle but functionally crucial transitions. Future reliance mitigation (e.g., OmniParser/Screen2AX) is acknowledged but not integrated (Sec. 5).
- Benchmarking scope: While macOS results are strong, overall ScreenSpot-Pro accuracy lags SOTA multi-OS models (Table 3, overall column), and qualitative failures on engineering/creative apps indicate domain gaps (Sec. 4.4; Table 6). Claims of superiority are strongest on macOS subsets and internal tests; broader parity is not yet achieved.
- Agentic evaluation: The agent study shows generally low success on input tasks across models (max 12.5%) and does not integrate the authors’ GUIRILLA-SEE models in a full agent loop with action execution, limiting end-to-end validation (Sec. 4.5).
- LLM reliance and reproducibility: Safety/ordering/input generation depends on GPT-4 (Sec. 3.1), which is closed and may affect reproducibility and cost. Although a deterministic mode exists, its performance/safety trade-offs are only summarized (App. A.7.2), not extensively benchmarked.
- Dataset balance and coverage: Heavy prevalence of button clicks (Fig. 5a) and limited coverage of icon-dense professional UIs (Sec. 4.4) may bias training. The strong macOS focus limits generality without explicit cross-OS crawling, though transfer results are promising (Sec. 4.3).

**Questions:**

- Coverage and complexity: How many tasks require multi-step reasoning or multiple actions? Could the authors quantify graph coverage completeness per app (e.g., proportion of reachable AX elements visited) and the distribution of graph depths beyond the average (Sec. 3.2)?
- Action space: Do you plan to support scroll, drag, right-click, and keyboard shortcuts? What changes are needed in the crawler/action schema and safety handlers to incorporate these?
- Heuristic thresholds: How sensitive are results to the “significant change” threshold of ±10 AX elements (Sec. 3.2)? Have you evaluated adaptive thresholds or visual change metrics to better capture state transitions?
- Deterministic vs. LLM-guided exploration: Can you provide a quantitative comparison of safety incidents, duplicate rates, and coverage between deterministic and GPT-guided modes across a larger app set (beyond App. A.7.2 examples)?
- AX noise mitigation: Beyond handlers, have you tried vision-first proposals with AX alignment (e.g., using OmniParser or Screen2AX) during crawling time? Could this reduce failure on creative/engineering tools identified in Sec. 4.4?
- Agent evaluation: Why weren’t GUIRILLA-SEE models evaluated in a closed-loop agent setting on GUIRILLA-TASK (agentic) akin to Sec. 4.5? If attempted, what were the bottlenecks (e.g., action APIs, decoding stability)?
- Data licensing and compliance: What is the legal stance for releasing full-desktop screenshots of third-party apps and their UI assets (icons, fonts)? Are there app categories excluded due to licensing? How will takedown requests be handled?
- Generalization: What’s the plan to extend GUIRILLA to Windows/Linux (Sec. 5)? Which handlers are platform-agnostic, and what new handlers are anticipated for those OSes (e.g., different menu/window conventions)?
- Cost and throughput: What were the LLM costs for the three agents per app, and how do costs scale with depth/time limits (Sec. 3.1; A.2)? Any batching or caching strategies to reduce cost?

**Details Of Ethics Concerns:**

- Legal/compliance: Full-desktop screenshots and UI assets from proprietary macOS apps may implicate copyright or App Store/EULA restrictions. The paper should detail consent, licensing, filtering, and takedown policies, and confirm adherence to platform terms during large-scale collection (Sec. 4.1, 5.1).

---

### Official Review · Reviewer_REtW · 2025-11-01

**Soundness:** 2
**Presentation:** 1
**Contribution:** 2
**Rating:** 2
**Confidence:** 4

**Summary:**

This work introduces a  GUIRILLA completely automated framework that can crawl macOS desktop applications and construct desktop application graphs via the native accessibility API and generate function-centric tasks. They funnel this framework with three GPT-4 based agents that order interactions, generate inputs, and refine task descriptions. Using this framework the authors ahs benerated a large dataset GUIRILLA‑TASK with 27k tasks spanned across 1.1k applications and 6.8k unique screens. They also manually verify a small sample (~5%) of dataset created using similar framework and release it as GUIRILLA‑GOLD (a test dataset).  Using the GUIRILLA‑TASK dataset, they fine0tuned 3 vision and language models (based on Florence-2-large 0.7B, Qwen2.5VL 3B and 7B) and benchamrk on their test dataset as well as ScreenSpot series benchmarks. The results demonstrate that with much less than compared to other models (that are fine-tuned on both human and synthetically generated) they perform on-par or better on the benchmarks.

**Strengths:**

1. Leveraging the macOS Accessibility API for systematic UI exploration and automating this could tackle the bottleneck of generating such large datasets that can push the boundaries of CUA performance on desktop applications.
2. This work identifies and address the  gap in CUA datasets in macOS desktop based applications.
3. Releasing the framework, code and the datasets created using the framework. If they are as impactful as described this could be very helpful for CUA research.
4. Privacy, security, and responsible aspects of dataset collection are handled properly ( providing concrete safeguards - sandboxed collection, local‑only operation).
5. Fine-tuned models on GUIRILLA‑TASK (completely trained on macOS applications) can generalise to other OS/Web on grounding tasks.

**Weaknesses:**

1. The core technical contribution is more on the engineering side and no new algorithmic/empirical contribution is made. Authors could have explored RL fientuning with small gold dataset to push the performance even further.
2. Many prior works have explored the usefulness of  accessibility APIs for data collection on certain level so this is not entirely new.
3. No comparison with closed-source models at least in Table2 to understand the difficultly of the test-set and how well off the shelf large close-source models perform on this task.
4. Like the previous point both experiments and empirical analysis is weak. Table3 is missing base model performance details (how does the base Qwen2.5VL 3/7B perform on this task). I believe the numbers on Screenspov2 shows the finetined GUIrilla-See models have better than base but that is not the case for ScreenSpotPro.
5. The quality analysis of the data shows questionable accuracy of (AX quality) ~40 % of elements have incorrect role/description labels, which can limit the usefulness of the raw accessibility trees for downstream models. Mitigation strategies (e.g., vision‑based correction) or what can be further done to make this framework and pipeline effective is not clear.

**Questions:**

Questions:

1. Qwen2.5VL models can be used with default prompt vs. agentic prompt for grounding and other tasks. Which one was used here for evaluation and fine-tuning?
2. How does the distribution of GOLD compare to TASK data?
3. Missing numbers on more challenging recently released datasets like UI-Vision, that not only test grounding accuracy but also action prediction.


Minor suggestions:

The formatting of citations is wrong. Use \citep instead of \cite.

In Table3 and other tables please organize results by model family or size or performance. For better readability and comparison of model performance.

Many sections could be written better or explained better.
UI-Vision: A Desktop-centric GUI Benchmark for Visual Perception and Interaction. https://arxiv.org/abs/2503.15661

---

### Official Review · Reviewer_FJf9 · 2025-11-03

**Soundness:** 3
**Presentation:** 2
**Contribution:** 3
**Rating:** 2
**Confidence:** 4

**Summary:**

This paper presents GUIRILLA, a scalable framework for automated desktop GUI exploration, specifically targeting the macOS ecosystem — a domain largely neglected in current GUI automation research. The framework uses native Accessibility APIs combined with GPT-4-based agents to systematically explore applications, build hierarchical GUI graphs, and synthesize functionally grounded tasks. Using this pipeline, the authors construct the GUIRILLA-TASK dataset containing 27,171 tasks across 1,108 applications, and release corresponding benchmarks and vision-language models. Empirical results demonstrate that fine-tuning on GUIRILLA-TASK significantly improves grounding performance on ScreenSpot-Pro, achieving comparable or better results than multi-OS baselines (UI-TARS, OS-Atlas) while using 97% less data.

**Strengths:**

1. The paper introduces the first open-source macOS-focused automated crawler that integrates Accessibility APIs and LLM-based reasoning agents. Its design for full-screen, multi-window exploration represents a substantial engineering and methodological advancement over prior datasets that rely on manual single-window annotation.

2. GUIRILLA-SEE models achieve performance on par with or better than large-scale baselines trained on up to 300× more data, demonstrating that function-oriented task design and structured GUI graphs significantly improve grounding quality and data utilization.

3. The release of GUIRILLA-TASK, GUIRILLA-GOLD, MACAPPTREE, and the full framework code provides the research community with a reproducible, extensible benchmark suite, enabling future exploration of GUI autonomy and accessibility technologies.

**Weaknesses:**

I am concerned about the results reported in Table 3. First, several grounding models achieve strong results on SSPRO — for example, Aguvis (I excluded models released within six months of the ICLR submission deadline) attains nearly 39.5% accuracy on SSPRO; why do the authors not mention these open‑source models? Second, I note that qwen2.5‑vl‑3b‑instruct and qwen2.5‑vl‑7b‑instruct achieve 23.9% and 29.0% accuracy on SSPRO, respectively, yet after further training reported by GUIrilla‑see they reach only 19.17% and 23.66% — a clear drop. This suggests the training data may be of low quality and may have harmed the base models' performance.

Also, the authors claim "less data, similar performance" as an advantage. However, although they deployed their automated crawler on 12,298 macOS applications, the final dataset includes tasks from only 23 app genres — a very small proportion of the original data. This raises concerns about whether the proposed framework has inherent flaws or is insufficiently optimized to capture useful data.

**Questions:**

1. The authors said they deployed automated crawler on 12298 macOS applications, yet the final dataset only contains tasks across 23 app generes -- why nearly 99% of the applications are dropped? I am worried that such a high filtering rate will limit the scaling potential of the dataset.
2. What reasons do the authors give for considering "AX Quality" unsatisfactory?

---

### Official Review · Reviewer_Rv2K · 2025-11-10

**Soundness:** 2
**Presentation:** 3
**Contribution:** 2
**Rating:** 4
**Confidence:** 3

**Summary:**

The current paper proposes a framework for automatic collection of multi-app full-desktop trajectories for GUI agents. The proposed method is used to collect a large dataset of tasks and associated trajectories across more than 1000 MacOS apps. The dataset is then used to train GUI agents, showing improvement on various downstream tasks. A manually curated dataset is also released in addition to the automatically generated one.

**Strengths:**

As manual annotation and collection of such data is not scalable, this is a very relevant topic now, in the era of training GUI agents.

In addition to the dataset, the authors fine-tune models on it showing improvements on downstream grounding and agentic tasks.

The graph structure where "edges" represent functional tasks is interesting, and different from the more common choice of having actions on edges.

**Weaknesses:**

The most value this paper brings is in its well-engineered platform-specific solution, and its applied impact. All the decisions in the GUIrilla crawler are very specific to MacOS. This has a limited fundamental impact, and it might be better communicated in the context of an workshop. (This is not a weakness, but maybe an argument for this paper to be redirected to a more applied audience.)

**Questions:**

1.  How important is the performance of the LLM behind the agents (input, order and login, task)? Understanding the cost of such a dataset collection effort.
2.  What is "graph depth"? Is it really a graph or a tree?

---

### Note · Authors · 2025-11-20

**Comment:**

Thanks for all the reviews and recommendations. We have seriously taken into review everything said here and reconsidered, as it is indeed more focused and practical work - we will apply it for a workshop submission.

**Withdrawal Confirmation:**

I have read and agree with the venue's withdrawal policy on behalf of myself and my co-authors.